**Data Availability Statement:** The data used for this study is publicly available from measure DHS website accessible after filling measure DHS data

# Determinants of maternal high-risk fertility behaviors and its correlation with child stunting and anemia in the East Africa region: A pooled analysis of nine East African countries

**Koku Sisay Tamirat** [ID]*, **Getayeneh Antehunegn Tesema, Zemenu Tadesse Tessema** [ID]

Department of Epidemiology and Biostatistics, Institute of Public Health, College of Medicine and Health Sciences, University of Gondar, Gondar, Ethiopia

* kokusisay23@gmail.com

## Abstract

### Background

In low-income nations, high-risk fertility behavior is a prevalent public health concern that can be ascribed to unmet family planning needs, child marriage, and a weak health system. As a result, this study aimed to determine the factors that influence high-risk fertility behavior and its impact on child stunting and anemia.

### Method

This study relied on secondary data sources from recent demography and health surveys of nine east African countries. Relevant data were extracted from Kids Record (KR) files and appended for the final analysis; 31,873 mother-child pairs were included in the final analysis. The mixed-effect logistic regression model (fixed and random effects) was used to describe the determinants of high-risk fertility behavior (HRFB) and its correlation with child stunting and anemia.

### Result

According to the pooled study about 57.6% (95% CI: 57.7 to 58.2) of women had at least one high-risk fertility behavior, with major disparities found across countries and women's residences. Women who lived in rural areas, had healthcare access challenges, had a history of abortion, lived in better socio-economic conditions, and had antenatal care follow-up were more likely to engage in high-risk fertility practices. Consequently, Young maternal age at first birth (<18), narrow birth intervals, and high birth orders were HRFBs associated with an increased occurrences of child stunting and anemia.

### Conclusion

This study revealed that the magnitude of high-risk fertility behavior was higher in east Africa region. The finding of this study underscores that interventions focused on health education

request form. The data are third-party data and not collected by the authors. The authors had no special access privileges when accessing the data. The data can be accessed through the website: https://dhsprogram.com/data/dataset_admin/index.cfm.

**Funding:** The author(s) received no specific funding for this work.

**Competing interests:** The authors have declared that no competing interests exist.

**Abbreviations:** ANC, Antenatal Care; AOR, Adjusted Odds Ratio; CI, Confidence Interval; CS, Cesarean Section; DHS, Demographic and Health Survey; EA, Enumeration Area; E, East; DHS, Ethiopian Demographic and Health Survey; ICC, Intra Class Correlation; IUD, Intrauterine Device; KR, Kids Record; LLR, Likelihood Ratio; MOR, Median Odds Ratio; N, North; PCV, Proportion of Cluster Variance; SE, Standard Error; SGD, Sustainable Development Goal; SSA, Sub-Saharan Africa.

and behavioral change of women, and improvement of maternal healthcare access would be helpful to avert risky fertility behaviors. In brief, encouraging contraceptive utilization and creating awareness about birth spacing among reproductive-age women would be more helpful. Meanwhile, frequent nutritional screening and early intervention of children born from women who had high-risk fertility characteristics are mandatory to reduce the burden of chronic malnutrition.

## Introduction

Rapid population growth has been observed in developing countries including Sub-Saharan African countries, with an estimated population of 1.2 billion by 2025 [1]. The total fertility rate is declining globally, but it is decreasing more slowly in SSA, with a total fertility rate of 4.69 children per woman in 2018, down from 5.37 in 2008. Despite declining natural resources, a lack of infrastructures such as housing, schools, and health facilities, and increased unemployment, most African countries lack a demography and population policy to control or monitor fertility rates [2].

Women's high-risk fertility habits, which are defined by narrow birth intervals, high birth order, and younger maternal age at birth, have been linked to negative health outcomes for both the mother and the child [3–5]. Due to increased family planning use, expansion of women's education, and economic trends, pooled decomposition analysis revealed that high-risk fertility behavior was decreased over decades. High-risk fertility behavior (HRFB) is linked to an increased risk of infant mortality, chronic malnutrition, and adverse birth outcomes such as stillbirth, prematurity, and low birth weight, according to research findings [6–11]. The nations of the East African region share many of the same socio-demographic and cultural characteristics. Maternal and child mortality remains high in this area, owing to risky fertility behaviors, the cultural taboo against contraception use, and insufficient health infrastructures. HRFBs are also common in the area due to child marriage, rape, and harmful sexual behaviors in elementary school [12–15].

Furthermore, nutritional problems among children under the age of five are common, as evidenced by the magnitude of stunting (36.7%) and anemia (60%) [16, 17]. Maternal HRFBs was one of the major contributors to infant malnutrition. For example, children who were born from women with high-risk fertility behavior had 40 percent and 43 percent more likely to suffer from stunting and anemia, respectively [16, 17]. As a result, a better understanding of the factors linked to risky fertility behavior and its consequences for child malnutrition may aid in the development of interventions. HRFB is more prevalent in low-income countries due to widespread poverty, a lack of basic health services, and early child marriage. Whilst interventions are challenging due to a lack of information about the magnitude and determinants of high-risk fertility behavior in the East Africa region.

Thus, this study aimed to discover factors that influence high-risk fertility behaviors and associations with child stunting and anemia. As maternal and child health is at the top of the region's agenda, the findings of this study may help to incorporate efforts at the Intergovernmental Authority for Development (IGAD) and African Union level.

## Method

### Data sources

This study was based on the secondary data from nine East African Demography and Health the most recent Survey (Burundi, Ethiopia, Malawi, Mozambique, Rwanda, Tanzania, Uganda,

Zimbabwe, and Madagascar) with the analysis period ranged from July 1–30, 2020. The appended datasets of countries were used to estimate the magnitude of high-risk fertility behavior and its effects among reproductive-age women. We included women in this study who had given birth in the five years before the survey and had a child under the age of five.

We used Kids Record (KR) files, which contain information about women and children, for this specific research. In terms of data extraction, we took women who were married and had completed data for the main variables, as well as children's anthropometric measurements. The data includes socioeconomic, reproductive health, and infant traits such as height for age and hemoglobin level. After data cleaning, the final sample size was 31,873 mothers-children pair who were included in the final analysis. To select study participants in each enumeration region, the DHS used a two-stage stratified sampling technique. We combined data from nine DHS surveys conducted in East African countries, yielding a weighted sample of 31,873 women and children. The strategy is described in detail in the DHS methodology section [16].

## Variables of the study

**Outcome variables.** *Maternal health outcome*. For this study, maternal high-risk fertility behavior was the primary outcome variable which is defined based on several criteria's as follow;

- High-risk fertility behavior is the outcome of interest for women who gave birth, defined as women age at birth less than 18 or above 34 years or birth interval less than 24 months or high birth order were criteria used to define [16].

- Single high-risk fertility behavior: when a woman reported to had one high-risk fertility behavior the is either younger age less than 18 years, or older age above 34 years, or birth interval less than 24 months, or high-birth order (four and above) [3, 17–19].

- Multiple high-risk fertility behavior: when a woman had a combination of at least two above-mentioned behaviors [3, 17–19]. Unavoidable high-risk fertility behavior is defined as women whose age was between 18 and 34 years and first birth order [16, 17].

- Unavoidable HRFB: when first-order births between ages of 18 and 34 years in women not amenable to the interventions.

- Not in any high-risk category: when women don't have any risk fertility behavior

*Children health outcomes*. another objective of this study was to see the association between maternal risky fertility behaviors and chronic malnutrition and anemia in children.

- Height-for-age is a measure of linear growth retardation and cumulative growth deficits. Children whose height-for-age Z-score is below minus two standard deviations (-2 SD) from the median of the reference population are considered short for their age (stunted), or chronically undernourished.

- Children who are below minus three standard deviations (-3 SD) are considered severely stunted.

- Anemia is a disease condition marked by low levels of hemoglobin, often below 10g/dl after correction for altitude [17].

- Mildly anemia: when the level of levels of hemoglobin between 10.0 and 10.9 g/dl [17].

- Moderately anemia: when the level of levels of hemoglobin between 7.0 and 9.9 g/dl [17].

- Severe anemia: when the level levels of hemoglobin less than 7g/dl [17].

### Independent variables

Socio-demographic and maternal health services like age group, sex of household headed, women's educational status, husband's educational status, maternal occupation status, marital status, media exposure, wealth status, sex of the child, birth order, antenatal care visits, sources of family planning, postnatal care visit, place of delivery, birth attendants, and healthcare access problems were independent variables.

**Data management and analysis.** After extracting the variables based on literature, data from the nine East African countries were combined. To restore the representativeness of the survey and take sampling design into account when calculating standard errors and reliable estimates, the data were weighted using sampling weight, main sampling unit, and strata before any statistical analysis. STATA version 14 was used to perform cross-tabulations and summary statistics.

Using a forest plot, the overall magnitude of high-risk fertility behavior, stunting, and anemia was estimated with the 95 percent Confidence Interval (CI). The DHS data had a hierarchical structure for the determinant factors, which contradicts the classical logistic regression model's independence of observations and equal variance assumptions. As a result, children are nested within a cluster, and we anticipate that children in the same cluster will be more similar than children across the country. This means that advanced models should be used to account for the variability between clusters. As a result, a mixed effect logistic regression model was fitted (with both fixed and random effects). Standard logistic regression and Generalized Linear Mixed Models (GLMM) were used because the outcome variable was binary (presence or absence of high-risk fertility behavior in women, stunting, and anemia in children). Since the models were nested, the Intra-class Correlation Coefficient (ICC), Likelihood Ratio (LR) test, Median Odds Ratio (MOR), and deviance (-2LLR) values were used to compare and assess model fitness. It was decided to use the model with the lowest deviance. As a result, the mixed-effect logistic regression model fits the data the best. In the multivariable mixed-effect logistic regression model, variables with a p-value of less than 0.2 in the bivariable analysis were considered. The multivariable model used Adjusted Odds Ratios (AOR) with a 95 percent Confidence Interval (CI) and p-value 0.05 to declare major factors high-risk fertility behavior. A multivariable Generalized Linear Mixed Models (GLMM) model was also fitted to see the relationship between HRFB and infant stunting and anemia. The HRFB had a major impact on stunting and anemia, as measured by the AOR with 95 percent confidence intervals and variables with a p-value less than 0.05.

**Ethical clearance and consent to participate.** Measure DHS provided ethical clearance after filling out a request for data access form. The data used in this study is aggregated secondary data that is publicly accessible and does not contain any personal identifying information that can be related to study participants. The data was kept confidential in an anonymous manner.

## Result

### Socio-demographic characteristics

A total of 31,873 study participants were drawn from nine East African countries, with Ethiopia, Tanzania, Madagascar, Burundi, Malawi, and Zimbabwe accounting for 21.8%, 15.6%, 12 percent, 11.3%, 10.9%, and 10.3%, respectively. The median age of respondents was 29 years, with an IQR of 25 to 35, and half of them aged between 25 and 35 years. The majority (80.5%) of women came from rural areas, nearly one-third (32.2%) had no formal schooling, and 45.4 percent lived in poverty (Table 1).

**Reproductive history of women.** The majority of the participants (88.4%) were multiparous, almost two-thirds (65.6%) gave birth in the health facilities, and about 4.5 percent gave

**Table 1. Socio-demographic characteristics of reproductive age women in east Africa region.**

| Characteristics | Frequency | Percentage |
|---|---|---|
| **Country** | | |
| Burundi | 3,631 | 11.4 |
| Ethiopia | 6,935 | 21.8 |
| Malawi | 3,492 | 11 |
| Mozambique | 2,254 | 7.1 |
| Rwanda | 1,701 | 5.3 |
| Tanzania | 4,976 | 15.6 |
| Uganda | 1,760 | 5.5 |
| Zimbabwe | 3,313 | 10.4 |
| Madagascar | 3,811 | 12 |
| **Age of respondents** | | |
| 15–19 | 1,168 | 3.7 |
| 20–24 | 6,413 | 20.1 |
| 25–29 | 8,782 | 27.6 |
| 30–34 | 7,341 | 23 |
| 35–39 | 5,044 | 15.8 |
| 40–44 | 2,393 | 7.5 |
| 45–49 | 732 | 2.3 |
| **Residence** | | |
| Urban | 6,899 | 19.3 |
| Rural | 28,785 | 80.7 |
| **Women level of education** | | |
| No formal education | 10259 | 32.2 |
| Primary school | 14893 | 46.7 |
| Secondary school | 5930 | 18.6 |
| Diploma and higher | 791 | 2.5 |
| **Husband education** | | |
| No formal education | 8034 | 25.2 |
| Primary school | 15239 | 47.8 |
| Secondary school | 6977 | 21.9 |
| Diploma and higher | 1623 | 5.1 |
| **Wealth index** | | |
| Poor | 14465 | 45.4 |
| Middle | 5887 | 18.5 |
| Rich | 11521 | 36.1 |
| **Household head** | | |
| Male | 26892 | 84.4 |
| Female | 4981 | 15.6 |
| **Media exposure** | | |
| Yes | 19653 | 60.3 |
| No | 12653 | 39.7 |
| Health insurance coverage | | |
| Yes | 2309 | 7.6 |
| No | 27863 | 92.4 |
| Women working condition | | |
| Yes | 22055 | 69.2 |
| No | 9818 | 30.8 |

(*Continued*)

**Table 1.** (Continued)

| Characteristics | Frequency | Percentage |
|---|---|---|
| Husband working condition | | |
| Yes | 30073 | 94.4 |
| No | 1800 | 5.6 |

birth by cesarean section. The majority (62.2%) had ANC follow-up, 21.2% of women had also postnatal follow-up, 40.3% of women had family planning details from the media, and 35.4% had discontinued family planning in the five years preceding the survey. More than two-thirds (68.2%) of women had trouble accessing healthcare due to a lack of resources, distance, permission, or companionship (Table 2).

## High-risk fertility behavior

The pooled analysis of this study indicated that 57.6% (95 percent CI: 57.7 to 58.2) of women had at least one high-risk fertility behavior, while 21.6 percent had multiple risk factors. The most common single high-risk fertility activity was higher birth order (45%), older age at birth (over 34 years) (15.7%), and birth period shorter than 24 months (15.6%). A combination of older women's age and higher birth order (age over 34 and birth order above 3) and a birth period less than 24 months and birth order above 3 accounted for 14.5 percent and 8.7% of women, respectively. Within the country, there was also variance, ranging from 66.59% in Uganda to 41% in Zimbabwe (Fig 1). Significant variations were also found between women from rural and urban areas, with risk differences of 17.68% and 7.21% for single and multiple high-risk fertility behaviors in the East Africa region, respectively (Fig 2) and (Table 3).

## Factors associated with high-risk fertility behavior

In the multivariable mixed-effect logistic regression model, mother and husband education levels, residence, country, wealth status, sex of household, place of delivery, delivered by CS, abortion, healthcare access problems, currently contraceptive utilization were variables correlated with high-risk fertility behaviors at a p-value of 0.05. In contrast to uneducated mothers, the chances of high-risk pregnancy activity were reduced by 41% (AOR = 0.59, 95% CI: 0.56 to 0.64), 68 percent (AOR = 0.32, 95% CI: 0.29 to 0.36), and 76% (AOR = 0.24, 95% CI: 0.19 to 0.29) for women who completed primary, secondary, and certificate and higher-level schooling. For those husbands who attended primary, secondary, diploma, and above level of education, the odds of high-risk fertility behaviors were reduced by 11% (AOR = 0.89, 95% CI: 0.83 to 0.95), 29% (AOR = 0.71, 95% CI: 0.65 to 0.78), and 25% (AOR: 0.75, 95% CI: 0.65 to 0.87) compared to low level of education, respectively.

Female-headed households had an 11% lower risk of high-risk fertility activity than male-headed households (AOR = 0.89, 95% CI: 0.83 to 0.95). Furthermore, rural mothers had a 1.26-fold higher probability of fertility activity than city mothers (AOR = 1.26, 95% CI: 1.17 to 1.36). Similarly, the chances of high-risk fertility activity were 1.10 times higher in wealthy women than in poor women (AOR = 1.10, 95%CI: 1.03 to 1.18). In addition, women with healthcare access challenges had a 10% higher risk of high-risk fertility activity than mothers who did not have such a history (AOR = 1.14, 95% CI: 1.08 to 1.20). Women who had terminated pregnancies had a 16 percent increased risk of high-risk fertility activity relative to those who had no such history (AOR = 1.16, 95% CI: 1.08 to 1.25). The chances of high-risk fertility activity were 1.51 times higher for mothers who gave birth at home (AOR = 1.51, 95% CI: 1.41 to1.61) than for mothers who gave birth in a hospital (AOR = 1.51, 95% CI: 1.41 to1.61).

**Table 2. Reproductive characteristics of child bearing women in East Africa region.**

| Characteristics | Frequency | Percentage |
|---|---|---|
| **Parity** | | |
| Primiparous | 3699 | 11.6 |
| Multiparous | 28174 | 88.4 |
| **Age at first birth** | | |
| Less than 18 | 1553 | 14.9 |
| 18–34 years | 25314 | 79.4 |
| Above 34 | 5006 | 15.7 |
| **Place of delivery** | | |
| Home | 10877 | 34.1 |
| Health facility | 20996 | 65.9 |
| **History of abortion** | | |
| Yes | 4336 | 13.6 |
| No | 27537 | 86.4 |
| **Current contraceptive use** | | |
| Yes | 18434 | 57.8 |
| No | 13439 | 42.2 |
| **The average size at birth** | | |
| Small | 5505 | 17.3 |
| Average | 15962 | 50.1 |
| Large | 10403 | 32.6 |
| **Delivered Cesarean section** | | |
| Yes | 1432 | 4.5 |
| No | 30391 | 95.5 |
| A faced healthcare access problem | | |
| Yes | 21748 | 68.2 |
| No | 10125 | 31.8 |
| ANC follow up | | |
| Yes | 19825 | 62.2 |
| No | 12048 | 37.8 |
| Postnatal follow-up | | |
| Yes | 6743 | 21.2 |
| No | 25130 | 78.8 |
| Sex of child | | |
| Male | 15980 | 50.1 |
| Female | 15893 | 49.9 |
| Discontinued contraceptive methods | | |
| Yes | 11,283 | 35.4 |
| No | 20,590 | 64.6 |
| Know the source of family planning | | |
| Yes | 14,184 | 55.5 |
| No | 17,689 | 44.5 |
| Had information about family planning | | |
| Yes | 12,853 | 40.3 |
| No | 19,020 | 59.7 |

Furthermore, women who had antenatal care follow-up with their recent baby had a 16% higher risk of delivering a healthy baby than those who did not (AOR = 1.16, 95% CI: 1.10 to 1.23). In contrast, mothers who were aware of the sources of family planning had an 11%

Proportion of high-risk fertility behavior among reproductive age women in East Africa

| country | Number of women with HRFB | Total population | | ES (95% CI) | % Weight |
|---|---|---|---|---|---|
| Burundi | 2172 | 3,631 | | 59.82 (58.22, 61.41) | 11.31 |
| Ethiopia | 4309 | 6,935 | | 62.13 (60.99, 63.28) | 22.06 |
| Malawi | 1733 | 3,492 | | 49.63 (47.97, 51.29) | 10.46 |
| Mozambique | 1367 | 2,254 | | 60.65 (58.63, 62.66) | 7.07 |
| Rwanda | 822 | 1,701 | | 48.32 (45.95, 50.70) | 5.10 |
| Tanzania | 3023 | 4,976 | | 60.75 (59.39, 62.11) | 15.62 |
| Uganda | 1172 | 1,760 | | 66.59 (64.39, 68.79) | 5.92 |
| Zimbabwe | 1357 | 3,313 | | 40.96 (39.29, 42.63) | 10.26 |
| Madagascar | 2391 | 3,811 | | 62.74 (61.20, 64.27) | 12.20 |
| Overall (I-squared = 98.9%, p = 0.000) | | | | 57.71 (57.17, 58.24) | 100.00 |

-68.8  0  68.8

**Fig 1. Forest plot of proportion of high-risk fertility behavior among reproductive-age women in East Africa countries.**

lower risk of high-risk fertility activity than those who were not aware of the sources of family planning (AOR = 0.89, 95% CI: 0.79 0.97). Women who gave birth by cesarean section had a 30% lower risk of high-risk fertility activity than women who gave birth vaginally (AOR = 0.70, 95% CI: 0.63 to 0.79). Similarly, women who were currently using contraception were reduced by 10% compared to those who were not currently using contraception (AOR = 0.90, 95% CI: 0.85 to 0.95). (Table 4).

**Association between maternal high-risk fertility behaviors and stunting and anemia in children.** To investigate the relationship between high-risk fertility activity and infant stunting and anemia, a mixed effect generalized linear mixed model (GLLM) was fitted. Thus, mothers under the age of 18 and over the age of 34, birth order greater than three, birth interval, and interactions of higher birth order and age greater than 34 years are associated with anemia stunting. Stunting was 1.55 (AOR = 1.55, 95% CI: 1.39 to 1.73), 1.33 (AOR = 1.33, 95% CI: 1.21 to 1.46) and 1.25 (AOR = 1.25, 95% CI: 1.18 to 1.32) times more likely in children born to mothers under the age of 18 at the time of birth, birth period less than 24 months, and birth order above three. Similarly, Similarly, an interactions of mother age over 34 and birth order greater than 3 was related to a 1.35 higher risk of infant stunting than those that did not have these characteristics (AOR = 1.35, 95 percent CI: 1.06 to 1.73). On other hand a mother's age at birth for 34 years is associated with a 25% lower risk of child stunting (AOR = 0.75, 95 percent CI: 0.60 to 0.95), compared to other age groups.

When the mother was less than 18 years old at the time of birth, the birth span was less than 24 months, and the birth order was greater than 3, the odds of infant anemia were 1.19 (AOR =

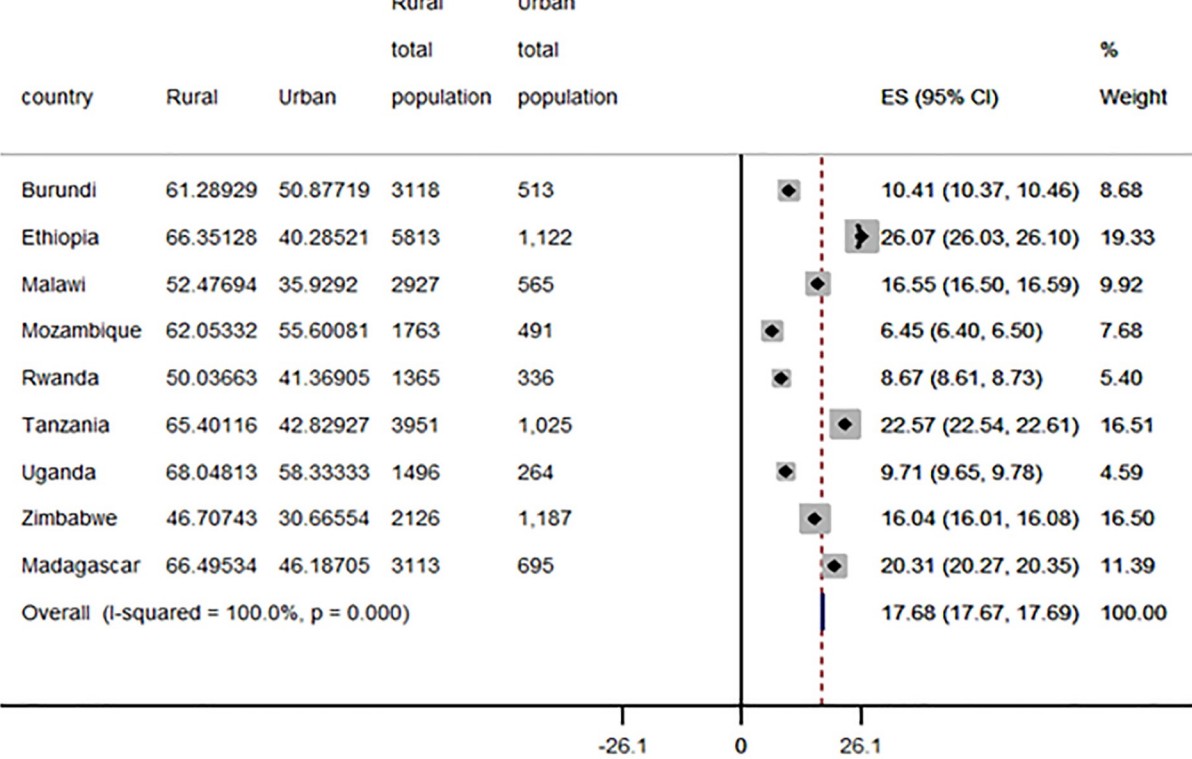

**Fig 2. Forest plot of risk differences of high-risk fertility behavior among between rural and urban reproductive-age women in East Africa countries.**

Table 3. High-risk fertility behavior of reproductive age women in east Africa region.

| Characteristics | Frequency | Percentage |
|---|---|---|
| **Any high-risk behavior** | | |
| Yes | 18346 | 57.6 |
| No | 13527 | 42.4 |
| **Single high-risk fertility behavior** | | |
| Age less than 18 years | 1553 | 4.9 |
| Age above 34 years | 5006 | 15.7 |
| Birth order above 3 | 14351 | 45 |
| The birth interval of less than 24 months | 4963 | 15.6 |
| **Multiple high-risk fertility behavior** | | |
| Age less than 18 years and birth interval less than 24 months | 121 | 0.4 |
| Age above 34 years and birth interval less than 24 months | 662 | 2.1 |
| Age above 34 years and birth order above 3 | 4619 | 14.5 |
| Birth interval less than 24 months and birth order above 3 | 2768 | 8.7 |
| Age above 34 years and birth interval less than 24 months, and birth order above 3 | 643 | 2 |
| **Unavoidable risk category** | | |
| First birth order and age of mother between 18 and 34 years | 4630 | 14.5 |
| **No high-risk fertility behavior** | 10,928 | 34.3 |

**Table 4. Factors associated with high-risk fertility behavior among women gave birth in east Africa region.**

| Characteristics | Odds ratio | | Characteristics | Crude 95%CI | Adjusted 95%CI |
|---|---|---|---|---|---|
| | Crude 95%CI | Adjusted 95%CI | Current contraceptive utilization | | |
| **Country** | | | Yes | 0.62(0.59 0.65) | 0.90(0.85 0.95)* |
| Burundi | 2.14(1.94 2.36) | 0.93(0.83 1.06) | No | 1 | 1 |
| Ethiopia | 2.32(2.13 2.53) | 0.73(0.65 0.83)* | **Sex of household** | | |
| Malawi | 1.40(1.27 1.55) | 0.87(0.78 0.97)* | Male | 1 | 1 |
| Mozambique | 2.23(2.0 2.49) | 1.06(0.92 1.21) | Female | 0.77(0.72 0.82) | 0.89(0.83 0.95)* |
| Rwanda | 1.35(1.20 1.52) | 0.77(0.67 0.88)* | **Faced healthcare access problem** | | |
| Tanzania | 2.20(2.01 2.41) | 1.09(0.98 1.22) | **Yes** | 1.56(1.48 1.63) | 1.14(1.08 1.20)* |
| Uganda | 2.85(2.52 3.23) | 1.68(1.46 1.93)* | **No** | 1 | 1 |
| Madagascar | 2.42(2.19 2.67) | 1.06(0.94 1.19) | **Delivered by CS** | | |
| Zimbabwe | 1 | 1 | Yes | 0.42(0.38 0.48) | 0.70(0.63 0.79)* |
| **Women level of education** | | | No | 1 | 1 |
| No formal education | 1 | 1 | **Residence** | | |
| Primary | 0.54(0.51 0.57) | 0.59(0.56 0.64)* | **Urban** | 1 | 1 |
| Secondary | 0.23(0.22 0.25) | 0.32(0.29 0.36)* | **Rural** | 2.20(2.07 2.33) | 1.26(1.17 1.36)* |
| Higher | 0.13(0.11 0.16) | 0.24(0.19 0.29)* | **Media exposure** | | |
| **Husband level of education** | | | Yes | 0.65(0.62 0.68) | 0.97(0.92 1.03) |
| No formal education | 1 | 1 | No | 1 | 1 |
| Primary | 0.68(0.64 0.72) | 0.89(0.83 0.95)* | **Postnatal care follow up** | | |
| Secondary | 0.34(0.32 0.36) | 0.71(0.65 0.78)* | Yes | 0.71(0.67 0.75) | 0.98(0.92 1.05) |
| Higher | 0.24(0.22 0.27) | 0.75(0.65 0.87)* | No | 1 | 1 |
| **Wealth status** | | | **ANC follow up** | | |
| Poor | 1 | 1 | **Yes** | 0.84(0.81 0.88) | 1.16(1.10 1.23)* |
| Middle | 0.86(0.80 0.91) | 1.03(0.96 1.10) | **No** | 1 | 1 |
| Rich | 0.54(0.51 0.57) | 1.10(1.03 1.18)* | **Know sources of family planning** | | |
| **Ever terminated pregnancy** | | | Yes | 0.71(0.68 0.74) | 0.89(0.79 0.97)* |
| Yes | 1.14(1.07 1.22) | 1.16(1.08 1.25)* | No | 1 | 1 |
| No | 1 | 1 | | | |
| **Place of delivery** | | | | | |
| Home | 2.04(1.94 2.14) | 1.51(1.41 1.61)* | | | |
| Health facility | 1 | 1 | | | |

1.19, 95 percent CI: 1.07 to 1.33), 1.12 (AOR = 1.12, 95 percent CI: 1.01 to 1.23), and 1.26 times higher than their counterparts. Women over 34 years old had a 28 percent lower risk of infant anemia than women of other ages (AOR = 0.72, 95 percent CI: 0.58 to 0.90) (Table 5).

## Discussion

This study intended to determine the pooled estimates of high-risk fertility behavior in East Africa countries. Thus, the pooled analysis revealed that 57.7% and 21.6% of women who gave birth had at least one and multiple high-risk fertility behavior. Of which, higher birth order, age above 34 at birth, and birth interval less than 24 months were the common single high-risk fertility behaviors. Moreover, significant variations were also observed among countries ranged from 41% in Zimbabwe to 66% in Uganda. Likewise, a significant difference was also observed between rural and urban mothers in terms of high-risk fertility behavior which accounted for 17% of risk differences (RD). The possible explanations for the observed variation might be child marriage practices, a high magnitude of unmet need for family planning,

**Table 5. Effect of high-risk fertility behavior on child chronic malnutrition and Anemia.**

| High-risk fertility behaviors | Stunting | | Crude OR | Adjusted OR | Anemia | | Crude OR | Adjusted OR |
|---|---|---|---|---|---|---|---|---|
| | Yes | No | | | Yes | No | | |
| Age less than 18 years | | | | | | | | |
| Yes | 711 | 842 | 1.35(1.21 1.50) | 1.55(1.39 1.73)* | 877 | 676 | 1.08(0.98 1.20) | 1.19(1.07 1.33)* |
| No | 18633 | 11687 | 1 | 1 | 16391 | 13929 | 1 | 1 |
| Age above 34 years | | | | | | | | |
| Yes | 2022 | 2984 | 1.07(1.01 1.14) | 0.75(0.60 0.95)* | 2660 | 2346 | 0.95(0.90 1.20) | 0.72(0.58 0.90)* |
| No | 10376 | 16491 | 1 | 1 | 14608 | 12259 | 1 | 1 |
| Birth interval less than 24 months | | | | | | | | |
| Yes | 2160 | 2803 | 1.26(1.18 1.34) | 1.33(1.21 1.46)* | 8161 | 2103 | 1.15(1.08 1.23) | 1.12(1.01 1.23)* |
| No | 10238 | 16672 | 1 | 1 | 9107 | 12502 | 1 | 1 |
| Birth order 4 and above | | | | | | | | |
| Yes | 5950 | 8401 | 1.21(1.16 1.27) | 1.25(1.18 1.32)* | 2860 | 6190 | 1.21(1.16 1.26) | 1.26(1.19 1.34)* |
| No | 6448 | 11074 | 1 | 1 | 14408 | 8415 | 1 | 1 |
| Age >34 years and Birth order >3 | | | | | | | | |
| Yes | 1907 | 2712 | 1.12(1.05 1.19) | 1.35(1.06 1.73)* | 2493 | 2126 | 0.99(0.93 1.06) | 1.19(0.95 1.50) |
| No | 10491 | 16763 | 1 | 1 | 14775 | 12479 | 1 | 1 |
| Age < 18 years & birth interval <24 months | | | | | | | | |
| Yes | 58 | 63 | 1.44(1.00 2.07) | 0.81(0.55 1.20) | 72 | 49 | 1.24(0.86 1.80) | 1.03(0.69 1.52) |
| No | 12340 | 19412 | 1 | 1 | 17196 | 14556 | 1 | 1 |
| Age >34 years & birth interval <24 months | | | | | | | | |
| Yes | 278 | 384 | 1.15(0.98 1.35) | 1.43(0.55 3.73) | 363 | 299 | 1.02(0.87 1.19) | 1.03(0.40 2.65) |
| No | 19091 | 12120 | 1 | 1 | 16905 | 14306 | 1 | 1 |
| Birth interval <24 months and birth order >3 | | | | | | | | |
| Yes | 1233 | 1535 | 1.30(1.20 1.41) | 0.92 (0.80 1.05) | 1658 | 1110 | 1.26(1.16 1.37) | 1.03(0.90 1.18) |
| No | 11165 | 17940 | 1 | 1 | 15610 | 13495 | 1 | 1 |
| Age above 34, birth order >3, and birth interval <24 months | | | | | | | | |
| Yes | 270 | 373 | 1.15(0.98 1.35) | 0.59(0.22 1.59) | 354 | 289 | 1.03(0.87 1.20) | 0.86(0.33 2.26) |
| No | 12128 | 19102 | 1 | 1 | 16914 | 14316 | 1 | 1 |

and bad cultural myths and beliefs to use family planning among women. In addition, most of the countries in Africa had no demography and population policy despite rapid population growth. In addition, these findings suggest that more interventions which focus on maternal health services like provision of family planning and counseling on reduction of risky fertility behaviors are very important.

Furthermore, there were also substantial risk variations in high-risk fertility activity between rural and urban areas. This may be explained by a lack of access to healthcare and family planning, suggesting that rural areas are the best place to participate to minimize maternal mortality, meet sustainable development goals, and achieve universal health coverage. This result was in line with results from Nepalese and Indian studies [20–22]. Women and husbands with some degree of schooling had lower risky-fertility behavior than women with no formal education, according to this report. This result was in line with the findings of other studies [20, 21, 23]. Women's awareness about the benefits of birth spacing and reproductive health attributes expanded as their educational levels rose. The effects of school reproductive clubs and the inclusion of fertility biology in the educational curriculum may also explain this. In contrast to male-headed households, female-headed households have a lower risk of high-risk fertility activity. This may be because women are responsible for both earning a living and caring for their children

Rich women, on the other hand, are more likely than poor women to participate in high-risk fertility activity. This may be because wealthier women (households) could want more children, which could contribute to risky fertility activity. This result was in line with previous research. Significant regional differences in high-risk fertility behavior were also discovered in this research. Women from Uganda had 1.68 times more likely to had high-risk fertility behavior than women from Zimbabwe, while women from Rwanda, Malawi, and Ethiopia had 23%, 13%, and 27% lower chances of high-risk fertility behavior than Zimbabwe, respectively. Regarding the place of delivery, women who gave birth at home had a greater high-risk fertility behavior than women who gave birth at a health facility [6, 7, 24, 25]. This result was in line with those of previous Ethiopian studies [26]. Immediate post-partum family planning programs, such as ICUD, were often available via health facilities. Integration and strengthening of family planning services with obstetrics services like IUCD insertion immediately after delivery.

Similarly, women who had ever terminated a pregnancy (abortion history) were more likely to engage in high-risk fertility activity than those who had not. This result was in line with previous results in Sub-Saharan Africa [9, 27]. Abortion was commonly associated with unwanted pregnancies with shorter birth periods and pregnancies at a young age, and it represented a lack of contraception use, which affected high-risk fertility. Women who had trouble accessing health services were often more likely to participate in high-risk fertility behaviors. This finding was consistent with previous research [28]. This may be because women who had trouble accessing health services used less family planning and received less ANC and postnatal care, resulting in shorter birth periods, births at an older age, and high birth orders. In addition, this study showed that mothers who received ANC during pregnancy had an increased risk of high-risk fertility activities relative to those who did not. This result contradicted previous research. This may be because women with shorter birth periods and conceptions at a later age are frequently high-risk and need regular monitoring and follow-up.

Another result of this study was that cesarean section delivery is associated with lower risky fertility activity as compared to vaginal delivery. This may be because frequent Cesarean section deliveries reduced the number of pregnancies due to the possibility of negative effects of repeated CS [29]. Similarly, during data collection, women who understood the origins of modern contraception and existing users were associated with lower risk fertility activity. This result was consistent with previous research [3, 5]. A strong understanding of contraceptive strategies and their use decreased unintended pregnancy and improved birth intervals.

This research, on the other hand, discovered a correlation between high-risk fertility activity and childhood chronic stunting and anemia. Thus, in the East African region, the prevalence of stunting and anemia was 38.9 percent and 54.2 percent, respectively, among those who gave birth in the five years preceding the study. In this report, the prevalence of chronic malnutrition (stunting) was lower than in India (45.1 percent) and Nepal (39.7 percent) [30]. This finding, however, was higher than that of Bangladesh (36%) and three disadvantaged east African countries (36.7%) [19, 31]. Furthermore, the incidence of infant anemia was lower than a study finding of 43.7 percent of Bangladeshis. This finding reflects that maternal fertility behaviors are also contributors to nutritional problems among children. Socio-cultural disparities, such as cultural taboos against some food products in Southeast Asia, maybe one reason. Another point to remember is the correlation between high-risk fertility and chronic malnutrition in children. As a result, women under the age of 18 at the time of birth were related to a higher risk of stunting and anemia. This result was in line with previous research. Birth age is often linked to social and health disadvantages and inequality. In comparison to other age classes, women over 34 years old at the time of birth have a lower risk of stunting and anemia.

Furthermore, children with a birth order greater than 3 and a birth interval of less than 24 months have a higher risk of stunting and anemia. This result was in line with previous

research. This may be attributed to a short birth period and a high birth order, which is related to intrauterine growth retardation in infants, maternal anemia, and maternal stress, both of which contribute to prematurity. Similarly, women with multiple high-risk fertility behaviors, such as age over 34 and high birth order of three or more, had a higher occurrence of high-risk fertility activity than those who did not. This finding was consistent with previous research. In general, this study found that high-risk fertility activity is widespread among East African reproductive-age women. Material high-risk fertility activity is linked to chronic malnutrition and anemia in children. This indicates that growing contraceptive use by women of childbearing age would help both the mother and the child's health.

For evidence-based approaches, this research has implications for reproductive-age women, healthcare planners, and policymakers. Furthermore, the results of this study revealed that amenable variables such as home delivery, educational status, wealth status, and contraceptive usage could be the target area for resolving the issues. In addition, factors such as schooling, residency, and family planning source have been described as strategies for reducing maternal and child mortality. However, there are some drawbacks to this research. First, the study's cross-sectional nature influenced the cause-effect relationship; second, health system characteristics were not assessed; and finally, the data in this study had recall bias issues, such as the number of months between births.

## Conclusion

This study revealed that the magnitude of high-risk fertility behavior was higher in the region. The finding of this study underscores that interventions focused on health education and behavioral change of women, and improvement of maternal healthcare access would be helpful to avert risky fertility behaviors. In brief, encouraging contraceptive utilization and creating awareness about birth spacing among reproductive-age women would be more helpful. Meanwhile, frequent nutritional screening and early intervention of children born from women who had high-risk fertility characteristics are mandatory to reduce the burden of chronic malnutrition.

## Acknowledgments

We would like to thank the Ethiopian Central Statistics Agency for providing us with all the relevant secondary data used in this study. Finally, we would like to thank all who directly or indirectly supported us.

## Author Contributions

**Conceptualization:** Koku Sisay Tamirat, Getayeneh Antehunegn Tesema, Zemenu Tadesse Tessema.

**Data curation:** Koku Sisay Tamirat, Getayeneh Antehunegn Tesema, Zemenu Tadesse Tessema.

**Formal analysis:** Koku Sisay Tamirat, Getayeneh Antehunegn Tesema, Zemenu Tadesse Tessema.

**Investigation:** Koku Sisay Tamirat, Getayeneh Antehunegn Tesema, Zemenu Tadesse Tessema.

**Methodology:** Koku Sisay Tamirat, Getayeneh Antehunegn Tesema, Zemenu Tadesse Tessema.

**Software:** Koku Sisay Tamirat, Getayeneh Antehunegn Tesema, Zemenu Tadesse Tessema.

**Supervision:** Koku Sisay Tamirat, Getayeneh Antehunegn Tesema, Zemenu Tadesse Tessema.

**Validation:** Koku Sisay Tamirat, Getayeneh Antehunegn Tesema, Zemenu Tadesse Tessema.

**Visualization:** Koku Sisay Tamirat, Getayeneh Antehunegn Tesema, Zemenu Tadesse Tessema.

**Writing – original draft:** Koku Sisay Tamirat, Getayeneh Antehunegn Tesema, Zemenu Tadesse Tessema.

**Writing – review & editing:** Koku Sisay Tamirat, Getayeneh Antehunegn Tesema, Zemenu Tadesse Tessema.

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
