## [Decision Letter · Decision Letter 0]

22 Feb 2021

PONE-D-20-27151

Determinants of maternal high-risk fertility behavior and its effects on stunting and anemia in the East Africa region: Pooled analysis of nine East African countries

PLOS ONE

Dear Dr. Tamirat,

Thank you for submitting your manuscript to PLOS ONE. After careful consideration, we feel that it has merit but does not fully meet PLOS ONE’s publication criteria as it currently stands. Therefore, we invite you to submit a revised version of the manuscript that addresses the points raised during the review process.

We look forward to receiving your revised manuscript.

Kind regards,

Frank T. Spradley

Academic Editor

PLOS ONE

2. Please include in your Methods section the date ranges of the DHS database analysed in the current study.

3.We note that you have indicated that data from this study are available upon request. PLOS only allows data to be available upon request if there are legal or ethical restrictions on sharing data publicly. For information on unacceptable data access restrictions, please see http://journals.plos.org/plosone/s/data-availability#loc-unacceptable-data-access-restrictions.

5. We note you have included a table to which you do not refer in the text of your manuscript. Please ensure that you refer to Table 3 in your text; if accepted, production will need this reference to link the reader to the Table.

Reviewers' comments:

Reviewer's Responses to Questions

**Comments to the Author**

1. Is the manuscript technically sound, and do the data support the conclusions?

Reviewer #1: Partly

2. Has the statistical analysis been performed appropriately and rigorously? 

Reviewer #1: Yes

3. Have the authors made all data underlying the findings in their manuscript fully available?

Reviewer #1: Yes

4. Is the manuscript presented in an intelligible fashion and written in standard English?

Reviewer #1: No

5. Review Comments to the Author

Reviewer #1: Please see below my comments.

1. Define abbreviations at first mention (e.g HRFB).

2. In the abstract, what do you mean by "Women and husband education"? please define the direction. Are you saying poor/low women and husband education?

3. The abstract's conclusion is not properly written and hard to follow. Please do not repeat the study result in the conclusion section.

4. The following statement in the methods section is confusing: "Whereas to see the relationship of risky behaviors with child chronic malnutrition and anemia, child hemoglobin level, and height for age measurements as dependent variables."

Which are/is the dependent variable(s)? High risk fertility behaviour? or chronic malnutrition? or anemia? or child hemoglobin level?

How were they defined? How were they expressed in the analysis? What is the diference between chronic malnutrition and height for age measurements in your study? I thought height for age measurements are used to measure chronic malnutrition.

5. Which are the exploratory variables? How were they defined? How were they expressed in the analysis?

6. How was statistical bias avoided?

In general, the sub-section: "Variables of the study" in the methods section should be extensively revised.

I would strongly recommend extensive grammar and punctuation editing.

6. PLOS authors have the option to publish the peer review history of their article (what does this mean?). If published, this will include your full peer review and any attached files.

Reviewer #1: No

---

## [Author Response · Author response to Decision Letter 0]

9 Apr 2021

Point by point response 

Manuscript title: Determinants of maternal high-risk fertility behaviors and its correlation with child stunting and anemia in the East Africa region: A Pooled Analysis of nine East African countries 

Manuscript ID: PONE-D-20-27151

Journal – PLOS ONE

Dear editor/reviewer

Dear all,

We would like to thank you for this constructive, building, and improvable comments on this manuscript that would improve the substance and content of the paper. We considered each comment and clarification questions of editors and reviewers on the document thoroughly. Our point-by-point responses for each comment and issues are described in detail on the following pages. Further, the details of changes were shown by track changes in the supplementary document attached.

Koku Sisay Tamirat 

Editor comments 

1. Please ensure that your manuscript meets PLOS ONE's style requirements, including those for file naming. The PLOS ONE style templates can be found at https://journals.plos.org/plosone/s/file?id=wjVg/PLOSOne_formatting_sample_main_body.pdf andhttps://journals.plos.org/plosone/s/file?id=ba62/PLOSOne_formatting_sample_title_authors_affiliations.pdf

• Author response: Thanks editors for your constructive comments, based on your comments we made all corrections according to Submission guidelines. 

2. Please include in your Methods section the date ranges of the DHS database analysed in the current study.

• Author response: Thanks editor for your comments for the purpose of this study we used secondary data sources from measure Demography and Health Survey (DHS) website with after fill the request form. The date of analysis for this study was from July 1-30, 2020. Mentioned in the method section, page 4, Line 88-89.

We note that you have indicated that data from this study are available upon request. PLOS only allows data to be available upon request if there are legal or ethical restrictions on sharing data publicly. For information on unacceptable data access restrictions, please see http://journals.plos.org/plosone/s/data-availability#loc-unacceptable-data-access-restrictions.

• Author response: Thanks reviewer for your constructive comments which are highly important to improve the quality of manuscript quality. 

• Author response: Thank you editor for your constructive comments. This study is further analysis of publicly available secondary data sources from measure DHS. Ethical clearance was obtained after filling an online data acquisition form of measure DHS. The request form available at www.measuredhs.com. 

• Author response: Corrected in the main document of the manuscript. Mentioned in the declaration section of the manuscript, page 17, line 357-358. 

 We note you have included a table to which you do not refer in the text of your manuscript. Please ensure that you refer to Table 3 in your text; if accepted, production will need this reference to link the reader to the Table.

• Author response: Corrected in the main document. Thank you editors for your insightful comments. it is already mentioned in the main document, page 9, line 188-189.

Reviewer 1: Is the manuscript technically sound, and do the data support the conclusions?

Reviewer #1: Partly

• Author response: The conclusion section corrected based on the results and objectives of the study.

Reviewer comments: Is the manuscript presented in an intelligible fashion and written in standard English?

 Reviewer #1: No

• Author response: The language proofread by all authors and other language exerts. Language errors edited and corrected in the main document of the manuscript.

Reviewer comments: Define abbreviations at first mention (e.g HRFB).

• Author response: Corrected in the main document of the manuscript HRFB stands to High-risk fertility behavior among reproductive age women. The detail mentioned in the introduction section and variables of the study section. Mentioned in the Page 3, line 63-64 and page 5, 103-113.

2. In the abstract, what do you mean by "Women and husband education"? please define the direction. Are you saying poor/low women and husband education? 

• Author response: Corrected as “In contrast to uneducated mothers, the chances of high-risk pregnancy activity were reduced by 41 % (AOR=0.59, 95 % CI: 0.56 to 0.64), 68 percent (AOR=0.32, 95 % CI: 0.29 to 0.36), and 76 % (AOR= 0.24, 95 % CI: 0.19 to 0.29) for women who completed primary, secondary, and certificate and higher level schooling. Those husband who attended primary, secondary, diploma and above level of education, the odds of high-risk fertility behaviors were reduced by 11 % (AOR=0.89, 95 % CI: 0.83 to 0.95), 29 % (AOR=0.71, 95 % CI: 0.65 to 0.78), and 25 % (AOR: 0.75, 95 % CI: 0.65 to 0.87) compared to low level of education, respectively. Corrected in the main document of the manuscript, page 9-10, Line 195-202.

Reviewer’s comments: The abstract's conclusion is not properly written and hard to follow. Please do not repeat the study result in the conclusion section.

• Author response: Thank you reviewer for your constructive comments. The abstract section revised and rephrased. Written as “Background: Low contraceptive utilization, child marriage, and a poor health system contributed to a high-risk fertility behavior in the East African region. As a result, this study aimed to establish determinants of high-risk fertility activity and their effect on child stunting and anemia. Method: This study relied on secondary data sources from recent demography and health surveys of nine east African countries. Relevant data were extracted from Kids Record (KR) files and appended for the final analysis; 31,873 mother-child pairs were included in the final analysis. The mixed-effect logistic regression model (fixed and random effects) was used to describe the determinants of high-risk fertility behavior (HRFB) and its correlation with child stunting and anemia. Result: According to the pooled study, 57.6% (95 % CI: 57.7 to 58.2) of women had at least one high-risk fertility behavior, with major a disparities found across countries and women's residences. High-risk fertility behaviors were more common among women of rural dwellers, faced healthcare access problems, history of abortion, better economic conditions, and had antenatal care follow-up. Consequently, younger women at first birth, narrow birth intervals, and high birth orders were HRFBs associated with an increased occurrence of child stunting and anemia. Conclusion: This study revealed that the magnitude of high-risk fertility behavior was higher in the region. The finding of this study underscores that interventions focused on health education and behavioral change of women, and improvement of maternal healthcare access would be helpful to avert risky fertility behaviors. In brief, encouraging contraceptive utilization and creating awareness about birth spacing among reproductive-age women would be more helpful. Meanwhile, frequent nutritional screening and early intervention of children born from women who had high risk fertility characteristics are mandatory to reduce the burden of chronic malnutrition.

4. The following statement in the methods section is confusing: "Whereas to see the relationship of risky behaviors with child chronic malnutrition and anemia, child hemoglobin level, and height for age measurements as dependent variables."

Which are/is the dependent variable(s)? High risk fertility behaviour? or chronic malnutrition? or anemia? or child hemoglobin level?

How were they defined? How were they expressed in the analysis? What is the diference between chronic malnutrition and height for age measurements in your study? I thought height for age measurements are used to measure chronic malnutrition.

• Author response: For this study there were more than one dependent variables. Thus, High-risk fertility behavior among reproductive women were the outcome variables for women. Mentioned as “High-risk fertility behavior is the outcome of interest for women who gave birth, defined as women age at birth less than 18 or above 34 years or birth interval less than 24 months or high birth order were criteria used to define the outcome of the interest.”

• Secondary outcomes of the study : Chronic malnutrition like Stunting and Anemia were also outcome variables for children to see any association between chronic malnutrition and High-risk fertility behavior among reproductive age women. Please note that for decision of HFRB and nutritional assessment used was for the recent child and birth. The details about the variables of the study mentioned in the method sections of the study. Page 5-6, Line 101-132.

Which are the exploratory variables? How were they defined? How were they expressed in the analysis?

• Author response: Corrected in the main document of the manuscript. Mentioned page 5-6, Line 101-132.

Reviewer comments: How was statistical bias avoided? In general, the sub-section: "Variables of the study" in the methods section should be extensively revised. I would strongly recommend extensive grammar and punctuation editing.

• Author response: Some of the strategies used to reduce bias in this study was using standardized definitions for classification of outcomes like HRFB, nutritional status like Anemia, stunting. Regarding the variables of the study we describe in detail in the method section of the manuscript. In addition we tried to address the language error through proof read by all authors.

---

## [Decision Letter · Decision Letter 1]

28 May 2021

PONE-D-20-27151R1

Determinants of maternal high-risk fertility behaviors and its correlation with child stunting and anemia in the East Africa region: A Pooled Analysis of nine East African countries

PLOS ONE

Dear Dr. Tamirat,

Thank you for submitting your manuscript to PLOS ONE. After careful consideration, we feel that it has merit but does not fully meet PLOS ONE’s publication criteria as it currently stands. Therefore, we invite you to submit a revised version of the manuscript that addresses the points raised during the review process.

We look forward to receiving your revised manuscript.

Kind regards,

Frank T. Spradley

Academic Editor

PLOS ONE

Journal Requirements:

Reviewers' comments:

Reviewer's Responses to Questions

**Comments to the Author**

1. If the authors have adequately addressed your comments raised in a previous round of review and you feel that this manuscript is now acceptable for publication, you may indicate that here to bypass the “Comments to the Author” section, enter your conflict of interest statement in the “Confidential to Editor” section, and submit your "Accept" recommendation.

Reviewer #1: All comments have been addressed

2. Is the manuscript technically sound, and do the data support the conclusions?

Reviewer #1: Yes

3. Has the statistical analysis been performed appropriately and rigorously? 

Reviewer #1: Yes

4. Have the authors made all data underlying the findings in their manuscript fully available?

Reviewer #1: Yes

5. Is the manuscript presented in an intelligible fashion and written in standard English?

Reviewer #1: Yes

6. Review Comments to the Author

Reviewer #1: Well-done on your revised manuscript. Your submission has greatly improved. However, there are still some grammar errors that makes some portions hard to understand. Here are some but I would recommend you further proofread your manuscript before final submission.

Revise your grammar:

Line 28, .......child marriage, and a poor health system contributes to high-risk fertility behaviour in the East African region.

Line 39, ......were common among women who live in rural areas, are unable to access healthcare, have history of abortion, have better economic conditions and had antenatal care follow-up.

I would recommend you use 'Young maternal age at first birth (<18)" rather than "younger women at first birth".

Line 115 - 116, what do you mean by ".....to see the relationship between risky behaviors with the child chronic malnutrition and anemia defined as follow," - this is hard to understand. Please revise the grammar.

What is the difference between "Unavoidable risk category" and "No high-risk fertility behavior"?.

7. PLOS authors have the option to publish the peer review history of their article (what does this mean?). If published, this will include your full peer review and any attached files.

Reviewer #1: No

---

## [Author Response · Author response to Decision Letter 1]

4 Jun 2021

Point by point response 

Manuscript title: Determinants of maternal high-risk fertility behaviors and its correlation with child stunting and anemia in the East Africa region: A Pooled Analysis of nine East African countries 

Manuscript ID: PONE-D-20-27151R1

Journal – PLOS ONE

Dear editor/reviewer

Dear all,

We would like to thank you for this constructive, building, and improvable comments on this manuscript that would improve the substance and content of the paper. We considered each comment and clarification questions of editors and reviewers on the document thoroughly. Our point-by-point responses for each comment and issues are described in detail on the following pages. Further, the details of changes were shown by track changes in the supplementary document attached.

Koku Sisay Tamirat 

Revise your grammar:

Line 28, .......child marriage, and a poor health system contributes to high-risk fertility behavior in the East African region.

Author response: Thanks reviewer for your constructive comments based on your suggestion grammatical errors corrected in the main document. Mentioned in Page 2, Line 28

Line 39, ……….were common among women who live in rural areas, are unable to access healthcare, have history of abortion, have better economic conditions and had antenatal care follow-up.

Author response: Thanks for your constructive comments it is already corrected in the main document abstract section. Page 2

I would recommend you use 'Young maternal age at first birth (<18)" rather than "younger women at first birth".

Author response: Thanks for your constructive comments it is already corrected in the main document abstract section. Page 2

Line 115 - 116, what do you mean by ".....to see the relationship between risky behaviors with the child chronic malnutrition and anemia defined as follow," - this is hard to understand. Please revise the grammar.

Author response: uthor response: Thanks for your constructive comments it is already corrected in the main document as “Children health outcomes: another objective of this study was to see the association between maternal risky fertility behaviors and chronic malnutrition and anemia in children”. 

What is the difference between "Unavoidable risk category" and "No high-risk fertility behavior"?.

Author response: Maternal health outcome: For this study, maternal high-risk fertility behavior was the primary outcome variable which is defined based on several criteria’s as follow; 

• High-risk fertility behavior is the outcome of interest for women who gave birth, defined as women age at birth less than 18 or above 34 years or birth interval less than 24 months or high birth order were criteria used to define [16]. 

• Single high-risk fertility behavior: when a woman reported to had one high-risk fertility behavior the is either younger age less than 18 years, or older age above 34 years, or birth interval less than 24 months, or high-birth order (four and above) [3, 17-19].

• Multiple high-risk fertility behavior: when a woman had a combination of at least two above-mentioned behaviors [3, 17-19]. Unavoidable high-risk fertility behavior is defined as women whose age was between 18 and 34 years and first birth order[16, 17]. 

• Unavoidable HRFB: when first-order births between ages of 18 and

34 years in women not amenable to the interventions. 

• Not in any high-risk category: when women don’t have any risk fertility behavior

---

## [Editor Report · Decision Letter 2]

14 Jun 2021

Determinants of maternal high-risk fertility behaviors and its correlation with child stunting and anemia in the East Africa region: A Pooled Analysis of nine East African countries

PONE-D-20-27151R2

Dear Dr. Tamirat,

We’re pleased to inform you that your manuscript has been judged scientifically suitable for publication and will be formally accepted for publication once it meets all outstanding technical requirements.

Kind regards,

Frank T. Spradley

Academic Editor

PLOS ONE

---

## [Editor Report · Acceptance letter]

16 Jun 2021

PONE-D-20-27151R2 

Determinants of maternal high-risk fertility behaviors and its correlation with child stunting and anemia in the East Africa region: A Pooled Analysis of nine East African countries 

Dear Dr. Tamirat:

I'm pleased to inform you that your manuscript has been deemed suitable for publication in PLOS ONE. Congratulations! Your manuscript is now with our production department. 

Kind regards, 

on behalf of

Dr. Frank T. Spradley 

Academic Editor

PLOS ONE